# Implementation of vascular surgery teleconsultation during the COVID-19 pandemic: Insights from the outpatient vascular clinics in a tertiary care hospital in Qatar

Hassan Al-Thani[1], Ahammed Mekkodathil[2], Ahmed Hussain[1], Ahmed Sharaf[1], Ahmed Sadek[1], Anas Aldakhl-Allah[1], Ahmed Awad[1], Nassar Al-Abdullah[1], Ahmad Zitoun[1], Jini Paul[1], Pushpalatha Pillai[1], Sara John[1], Ayman El-Menyar[2,3]*

1 Department of Surgery, Vascular Surgery, Hamad General Hospital (HGH), Doha, Qatar, 2 Department of Surgery, Vascular Surgery, Clinical Research, HGH, Doha, Qatar, 3 Department of Clinical Medicine, Weill Cornell Medical College, Doha, Qatar

* aymanco65@yahoo.com

## Abstract

### Background

The COVID-19 pandemic has sparked a surge in the use of virtual communication tools for delivering clinical services for many non-urgent medical needs allowing telehealth or telemedicine, to become an almost inevitable part of the patient care. However, most of patients with vascular disease may require face-to-face interaction and are at risk of worse outcomes if not managed in timely manner.

### Objective

We aimed to describe the utilization of telemedicine services in the outpatient vascular surgery clinics in a tertiary hospital.

### Methods

A retrospective analysis of data on all vascular outpatient encounters during 2019 and 2020 was conducted and compared to reflect the pattern of practice prior to and during the COVID-19 pandemic.

### Results

The study showed that 61% of the total patient encounters in 2020 were reported through teleconsultation. Females were the majority of patients who sought the virtual vascular care. Consultations for the new cases decreased from 29% to 26% whereas, the follow-up cases increased from 71% to 74% in 2020 (p = 0.001). The number of procedures performed in the vascular outpatient clinics decreased by 46% in 2020 when compared to 2019. This decrease in procedures was more evident in the duration from February 2020 to April 2020

**Data Availability Statement:** Data cannot be shared publicly because it includes potentially sensitive information, therefore access is limited to by request by the Ethical Review Board (IRB# MRC-01-21-430) at Hamad Medical Corporation (HMC) in Doha, Qatar. Data are available from the Medical Research Center (MRC) at HMC (contact via e-mail: mrchelpdesck@hamad.qa) for researchers who meet the criteria for access to confidential data.

**Funding:** The author(s) received no specific funding for this work.

**Competing interests:** The authors have declared that no competing interests exist.

in which the procedures decreased by 97%. The proportion of procedures represented 22.6% of the total encounters in 2019 and 10.5% of the encounters during 2020, (p = 0.001).

## Conclusions

Teleconsultation, along with supporting practice guidelines, can be used to maximize the efficiency of care in vascular surgery patients during the pandemic and beyond. Adoption of the 'hybrid care' which combines both virtual and in-person services as an ongoing practice requires evidence obtained through audits and studies on patients and healthcare providers levels. It is essential to establish a clear practice that ensures patient's needs.

## Introduction

The Coronavirus Disease 2019 (COVID-19) caused by Severe Acute Respiratory Syndrome Coronavirus 2 (SARS-CoV-2) was declared as a global pandemic by the World Health Organization (WHO) in March 2020 [1, 2]. The pandemic had an impact across all aspects of healthcare systems around the world with delays and disruption of services that are still being contended with today. However, the pandemic also sparked a surge in the interest and use of virtual means of delivering clinical services. The delivery of clinical services through phone calls, videos or video-calls and/or text has become a new normal for non-urgent medical needs [3]. Telehealth, telemedicine or teleconsultation is referred as the use of telecommunication technologies for the distribution of health-related services such as long-distance clinical health care, patient and professional health-related education, public health, and health administration. While it has been in existence since the late 1950s and early 1960s, it has undergone a transformation and has been re-centered as a common practice due to the pandemic [4, 5]. Previous studies on telemedicine in vascular surgery have shown that commercially available hardware and software solutions provide secure virtual patient visits and have proven to be effective in the management of patients with chronic vascular diseases [6, 7]. Patients with vascular health issues like acute ischemia, stroke, bleeding, venous ulcers, and deep vein thrombosis who need vascular access procedures become at risk of worse outcomes if they are not managed in timely and appropriate manner. Synchronous telemedicine with point-of-care ultrasound was found to be effective in the evaluation of common vascular conditions [6]. In addition, telemedicine can be used to reach a larger population, reduce financial burden on patients especially by reducing patient travel time and cost, reduce environmental pollutant emissions and reduce the use of personal protective equipment (PPE) [6–8]. While the switch to telemedicine has accelerated due to the pandemic, it is important to be mindful of the social and legal parameters within the practice context.

During the COVID-19 pandemic, the Hamad Medical Corporation (HMC) in Qatar, in which Hamad General Hospital (HGH) is the only tertiary facility for vascular surgery service in the country, has provided around 90 percent of the regular services, including urgent consultations, via telemedicine (teleconsultation) [9]. This physician-led service allows patients to have a virtual consultation with their healthcare providers, receive medical advice, sick leave and get their medicine delivered to their doorstep. Audits and patient satisfaction on the use telemedicine remain crucial since many of the patients with vascular diseases necessitated face-to-face interactions. However, as discussed above, evidence suggests that the use of telemedicine is appropriate especially in pandemic situations, such as COVID-19. The aim of this study is to explore the pattern of utilization of telemedicine services in the vascular surgery

outpatient clinic before and during the COVID-19 era in a tertiary hospital in Qatar. The study findings may be relevant to improve the other patient care settings globally during an emergency pandemic.

## Methods

A retrospective analysis of data on all patient encounters from the 1st of January 2019 to 31st of December 2020 at the vascular surgery clinic of HGH was conducted. The study design was approved by the Medical Research Center (IRB# MRC-01-21-430), HMC, Doha, Qatar. The need for consent was waived as this was a retrospective analysis, and the data were anonymously collected.

Data were presented by gender, nationality (national vs. non-national), new cases, follow-up cases, procedures, physical consultation and teleconsultation (such as electronic, digital, Internet-based, or telephone-based communication for direct patient care). These variables were compared from 2019 and 2020 to reflect the data prior to and during the COVID-19 pandemic in the country. Procedures performed in the clinic mainly include sclerotherapy, duplex ultrasound scan and diabetic foot dressing. Vascular surgery services at HGH are summarized in Table 1.

Although a telemedicine/teleconsultation service at the vascular clinic at HGH was established prior to the start of 2020, its use significantly increased from March 2020. Virtual consultations were mainly offered over telephone to limit face-to-face interactions for non-urgent cases and for patient follow-up. This service could be accessed by dialing the toll-free number 16000, between 7am and 10pm any day of the week, to schedule an appointment [9]. Fig 1 provides a comparison of the virtual and in person encounters at the vascular surgery clinic in 2019 and 2020.

Fig 2 provides a flow-chart of triage and referrals at the vascular clinics. The COVID-19 data were obtained from Qatar Open Data portal by the Government of Qatar [10]. All teleconsultations were delivered to patients at no cost and medications were dispensed to the patient address via collaboration between the hospital pharmacy and post-office express service. All the information in the outpatient clinics was based on telephonic conversation.

Data were presented as numbers and percentages. Chi-square test was performed to examine the significance of differences in proportion between categorical variables under study (2019 *vs* 2020 era). A 2-tailed p-value less than 0.05 was considered statistically significant. In this study, a comparative analysis of patient encounters, patient characteristics and procedures performed were compared before and during the COVID-19 pandemic. The percentage change over the time was calculated to quantify the change from one number to another and express the change as an increase or decrease. Data analysis was carried out using SPSS version 18 (SPSS Inc., Chicago, Illinois).

## Results

During the study period of 2 years (2019–2020), a total of 23,894 patient encounters (n = 11,105 in 2019 and n = 12,789 in 2020) were recorded. In the year 2019, all consultations were physical while nearly 61% of the total encounters were through telemedicine services in 2020 (Table 2). Fig 3 summarizes the vascular patients' workload in the outpatient vascular surgery clinic during the pandemic. Most cases had varicose vein disease followed by peripheral artery disease.

Fig 4 demonstrates the dramatic drop in number of physical visits of patients (dropped by 95%) in the duration from February 2020 to April 2020. The same period showed that the telephone consultations increased more than 25 times.

**Table 1. Key vascular services at Hamad General Hospital.**

- **Vascular access for dialysis**:
  - Catheter access.
  - Arteriovenous fistula.
- **Dialysis peritoneal access**.
- **Vascular access for chemotherapy**:
  - Catheter access.
  - Implanted access.
- **Vascular access for nutritional therapies**:
  - Short-term access.
  - Long-term access.
  - Vascular access for Extracorporeal Membrane Oxygenation (ECMO).
- **Vascular surgery of neurovascular pathologies**:
  - Carotid endarterectomy (e.g. CVA).
  - Implanted devices for epilepsy.
- **Vascular surgery for atherosclerosis**:
  - Occlusive diseases; peripheral arterial disease, chronic mesenteric ischemia.
  - Aneurysmal diseases; abdominal aortic aneurysm, thoracic aortic aneurysm or others.
- **Vascular endovascular interventions**.
  - For both occlusive disease and aneurysms.
- **Major vascular injury procedures**.
- **Diabetic foot management**.
- **Venous surgery** i.e. procedures related to varicose veins.
- **Lymphatic disease management**.
- **Oncology- Vascular surgery** in MDT to support other surgical specialties for cancer cases.
- **Multi-disciplinary outpatient vascular clinic for hemodialysis** vascular access care with imaging services "One stop vascular shopping" to improve the quality of care and reduce waiting time for patients.
- **Network vascular healthcare** with all the HMC Hospitals to coordinate patients' care.
- **Training for healthcare professionals** (resident, nurses).
- Provide teaching activities and CME for healthcare professionals in vascular surgery and medicine.
- Vascular research.
- **Vascular quality assurance**.

Females were the majority of patients (62%) across the study. There were no significant differences in males and females accessed care in 2019 and 2020 (p = 0.07) (Table 2). However, there were significant differences in the proportion of Qatari nationals and non-nationals accessing the care each year. Qatari patient encounters decreased by 2% in 2020 and the proportion of non-Qatari patient encounters increased accordingly (2%) (p = 0.001). Of note, the Qatari population (citizens) represents only 15% of the total Qatar population. Further analysis revealed that the non-national males (74%) and non-national females (65%) were the majority among male and female groups. There was 4% percentage decrease among the Qatari females who accessed care in 2020 (p = 0.001) (Fig 5).

Of the total, 27% were new cases whereas 73% were follow-up cases. There was significant drop in the new cases and increase in the follow-up cases in 2020. New cases decreased from 29% to 26% whereas follow-up cases increased from 71% to 74% in 2020 (p = 0.001).

The number of procedures performed in the vascular clinics decreased by 46% in 2020 when compared to 2019. This decrease was prominent in the duration from February 2020 to April 2020 in which the procedures decreased by 97%. The proportion of procedures in the vascular clinic represented 22.6% of the total encounters in 2019 and 10.5% of the encounters during 2020 (p = 0.001) (Table 2). The main procedures performed in the study duration were

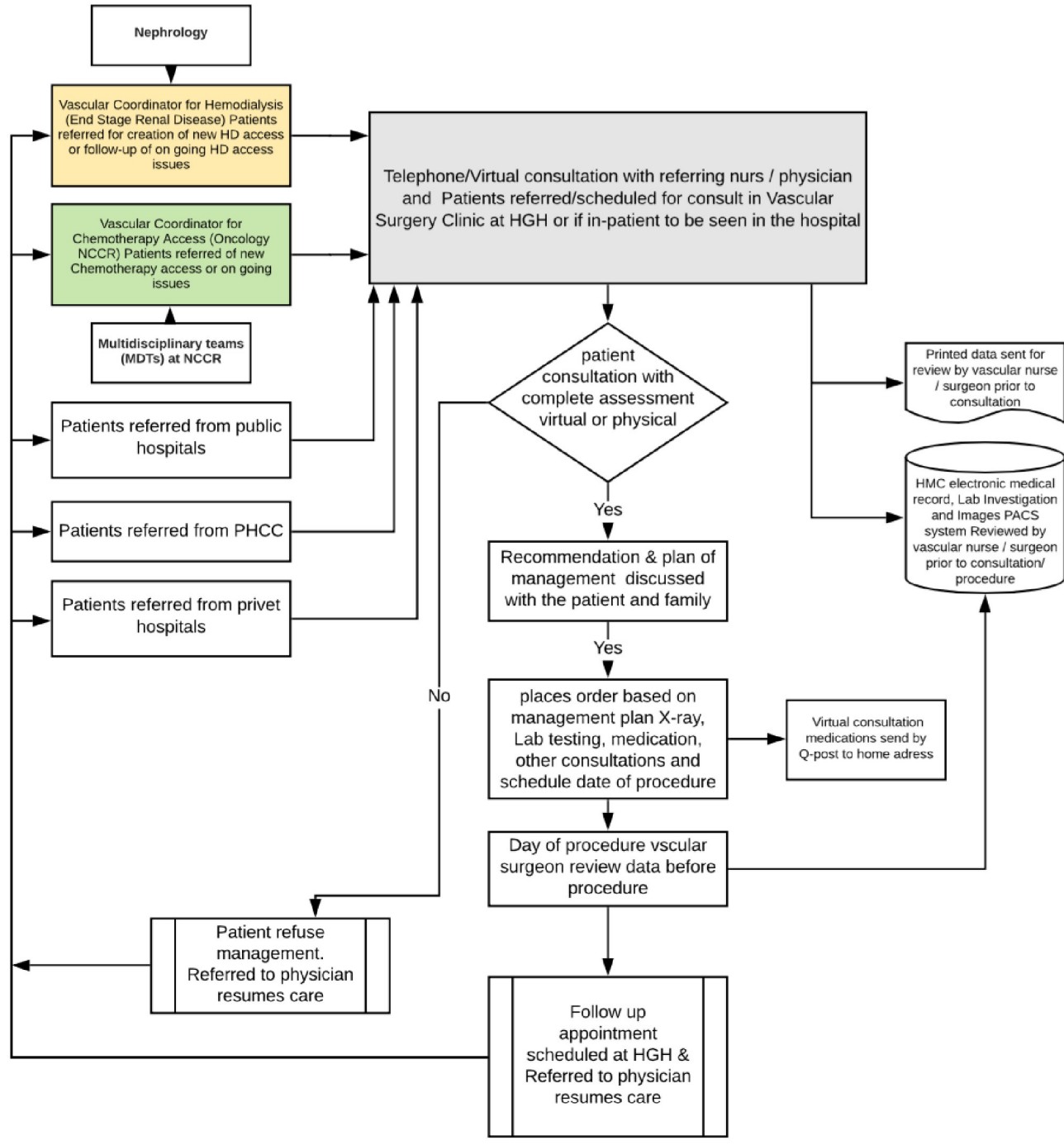

**Fig 1. Telephone/Virtual consultation in vascular surgery clinic at Hamad General Hospital.**

duplex ultrasound scan (38.1%), sclerotherapy (34.2%), and diabetic foot dressing (14.5%). Other procedures included foam sclerotherapy, cutaneous laser treatment, clip removal, permcath removal, surgical dressing, vascular ulcer Unna boot, suturing and Hickman removal. Notably, sclerotherapy (liquid or foam) decreased from 56% of the total procedures to 34% whereas duplex ultrasound scan and diabetic foot dressing increased significantly in 2020.

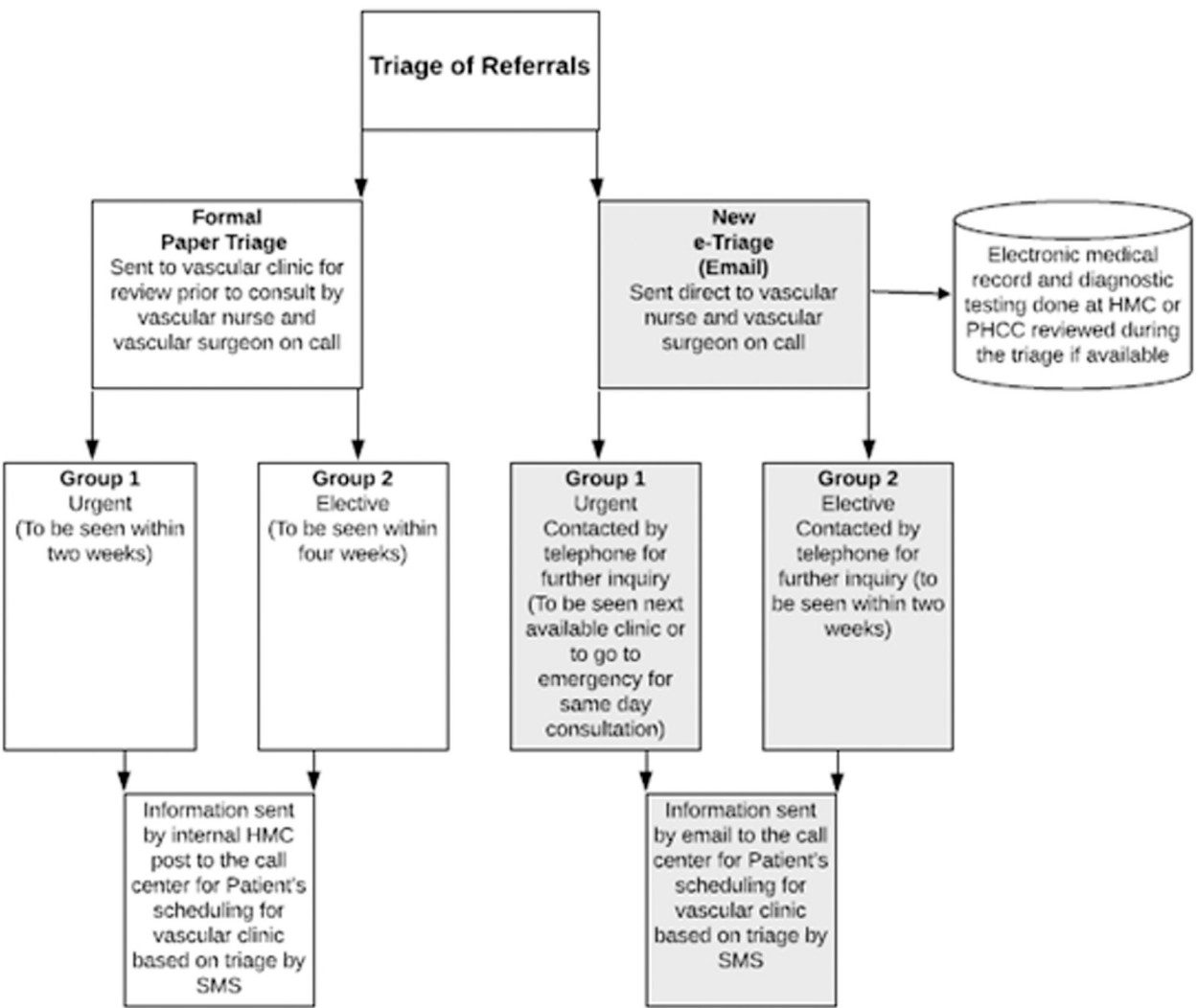

**Fig 2. Triage of referrals at the vascular surgery clinic at Hamad General Hospital.**

Sclerotherapy procedure was stopped during the COVID pandemic as illustrated in Fig 4. (459 cases in the first three months and only one case in September for bleeding varicose vein).

## Discussion

The present study from the vascular surgery outpatient clinic in the only tertiary care facility within Qatar demonstrated the increased use of telemedicine especially during the peak of COVID-19 pandemic. Our study demonstrated that 61% of the outpatient encounters at the vascular clinic was through teleconsultation in the year 2020 (COVID-19 era). Females were the majority of patients who accessed the virtual care in this period, and many of cases were follow-ups. Consultations for new vascular cases reduced significantly. The duration from February 2020 to April 2020 witnessed 95% drop in the physical visits of patients and more than 25 times increase in the telemedicine encounters. Similarly, there was 97% decrease in procedures performed in the vascular clinics during the COVID-19 period. During the peak of the pandemic, there was a shift in the multi-disciplinary outpatient vascular clinic for hemodialysis

**Table 2. Patients consultations, characteristics, and procedures at the vascular surgery outpatient clinic at Hamad General Hospital before and during the COVID-19 era (N = 23,984).**

|  | 2019 | 2020 | P-value |
|---|---|---|---|
| **Total consultations** | **11,105 (46.5%)** | **12,789 (53.5%)** | 0.001 for all |
| Physical consultations | 11,105 (100%) | 4965 (38.8%) * | |
| Telemedicine services | 0 | 7737 (60.5%) | |
| **Gender** | | | 0.07 for all |
| Male | 4249 (38.3%) | 4871 (38.1%) | |
| Female | 6856(61.7%) | 7918 (61.9%) | |
| **Nationality** | | | 0.001 for all |
| Nationals | 3608 (32.5%) | 3895 (30.5%) | |
| Non-nationals | 7497 (67.5%) | 8894 (69.5%) | |
| **Type of service** | | | 0.001 for all |
| New cases | 3171 (28.6%) | 3268 (25.6%) * | |
| Follow-up | 7934 (71.4%) | 9511 (74.4%) | |
| **Procedures done**: | **2506 (22.6%)** | **1346 (10.5%)** | 0.001 for all |
| Sclerotherapy (liquid & foam) | 1414 (56.4%) | 460 (34.2%) | |
| Duplex ultrasound scan | 455 (18.2%) | 513 (38.1%) | |
| Diabetic foot dressing | 320 (12.8%) | 195 (14.5%) | |
| PermCath removal | 98 (3.9%) | 28 (2.1%) | |
| Cutaneous laser treatment | 87(3.5%) | 17 (1.3%) | |
| Venous ulcer Unna boot | 74 (3.0%) | 62 (4.6%) | |
| Surgical dressing | 28 (1.1%) | 38 (2.8%) | |
| Suture removal | 14 (0.6%) | 29 (2.2%) | |
| Others** | 16 (0.5%) | 4 (0.2%) | |

*Valid percentages are used.

** Others include clip removal, suturing and Hickman removal.

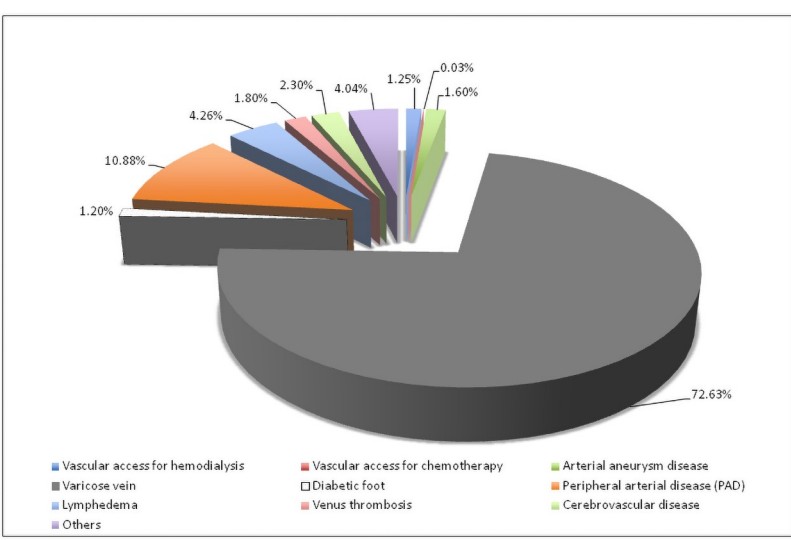

**Fig 3. Vascular patients' workload during the pandemic in the outpatient vascular surgery clinic during the pandemic.**

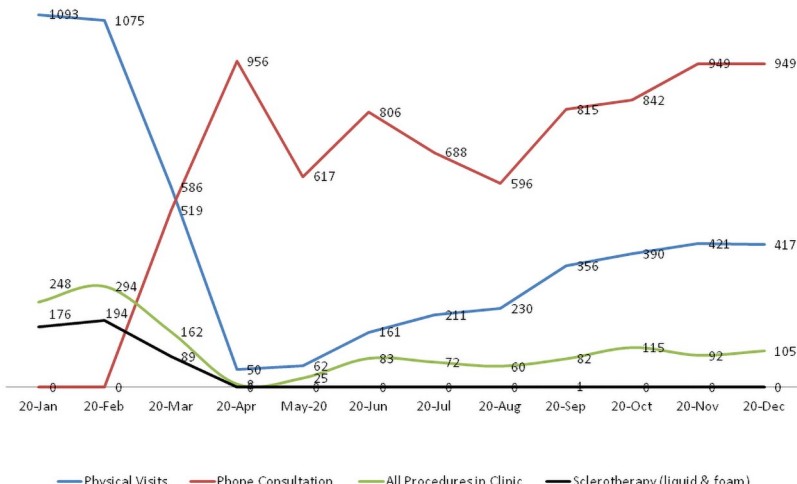

**Fig 4. Patient encounters and procedures done at vascular surgery outpatient clinic at Hamad General Hospital during the COVID-19 period.**

vascular access to inpatient consultation or direct consultation in the hemodialysis centers. This resulted in a significant drop in the number of physical visits to the vascular clinic for the hemodialysis patients.

On the other hand, patients for vascular access for chemotherapy rarely come to the vascular clinic even before the pandemic. The vascular coordinators for chemotherapy access at the national oncology center (NCCR) arrange referral for new chemotherapy access or ongoing issues with patients who need chemotherapy and then patients will be seen at the NCCR physically by the vascular team.

All the emergency vascular consultations are seen by the vascular surgeon on-call physically whether inpatient or in the emergency department. The pandemic did not impact on the overall number of consultations. On the other hand, the surgical procedures decreased from 2506 in 2019 to 1849 in 2020 due to the discontinuation of elective varicose vein surgery. Oncology-Vascular surgery Multidisciplinary Video Conferencing Meetings Calling using Microsoft Teams to support other surgical specialties for cancer cases take place twice weekly; one the uro-oncology Multidisciplinary and the other one in the sarcoma Multidisciplinary team.

The Primary Health Care Corporation (PHCC) and HMC had to be prepared to continue to address the needs of the 2.7 million people that make up Qatar's population along with managing the ongoing pandemic. The PHCC operates 27 primary health care centers across the country. HMC is the premier not-for-profit healthcare provider and has 9 hospitals under its umbrella including the only tertiary care hospital, HGH, which includes the vascular surgery clinics.

The healthcare system in Qatar has managed remarkable development in its relatively short history. The HMC began implementation of electronic medical records in 2011 using Cerner Millennium and by the end of 2017, all HMC hospitals; PHCC and Sidra Medicine implemented one Clinical Information System (CIS). In 2015, PricewaterhouseCoopers (PwC), a multinational audit, assurance and consultation firm, was invited to support the development of the Qatar National E-Health and Data Program (QNeDP) with the vision of creating a world-class, sustainable, integrated and secure National E-Health ecosystem. The Healthcare Information and Management Systems Society (HIMSS) recently announced that two HMC

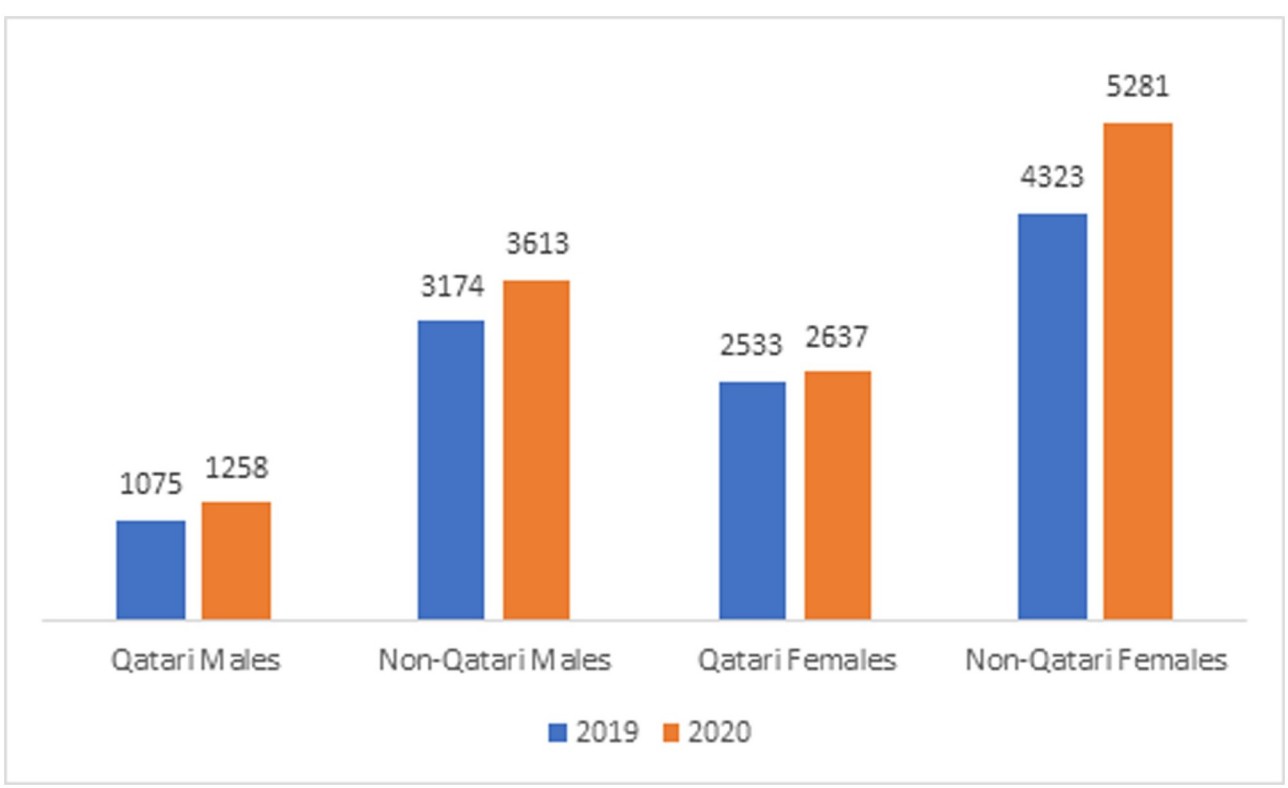

**Fig 5. Access of vascular surgery clinic at Hamad General Hospital by gender and nationality during the COVID-19.**

facilities, the Heart Hospital and the NCCCR, achieved the HIMSS Analytics Electronic Medical Record Adoption ModelSM (EMRAM) Stage 6 Distinction, an international benchmark for the use of advanced IT to improve patient care. In 2018, the patient portal, "MyHealth", service enabled individuals to register themselves and their children to access their health records online anytime, anywhere, including laboratory test results, medication details and medical appointments. This was crucial to the healthcare's system to continue providing services during the pandemic.

The first positive COVID-19 case in Qatar was confirmed on 29th February 2020, and the infection rate in the country has peaked in June 2020 (Fig 6).

The significant decrease in the physical consultations, corresponding increase in the teleconsultations and drop in procedures performed in the clinic reflect the public health measures adopted by the country to prevent spread of COVID-19, such as minimizing direct contact with patients for non-urgent care. Although there is a lack of studies on patient satisfaction, experts in this subject area reported that telemedicine is gaining popularity among vascular patients, especially among the follow-up patients. Therefore, new normal patient encounters utilizing telemedicine services may emerge in the post-COVID era, perhaps with a hybridization of virtual and in-person experiences across the full-continuum of care.

A study from the vascular surgery department in Seattle, USA demonstrated a significant change in all the normal vascular surgery practice during the COVID-19 period after adopting new practice guidelines [11]. The average weekly clinical volume and surgical volume decreased by 96.5% and 71.7%, respectively. Sixty percent of the inpatient consultations were attained as teleconsultation in which the patient was never evaluated in-person. The trainee surgical volume has also decreased by 86.4% for the vascular surgery fellow. In contrast, a

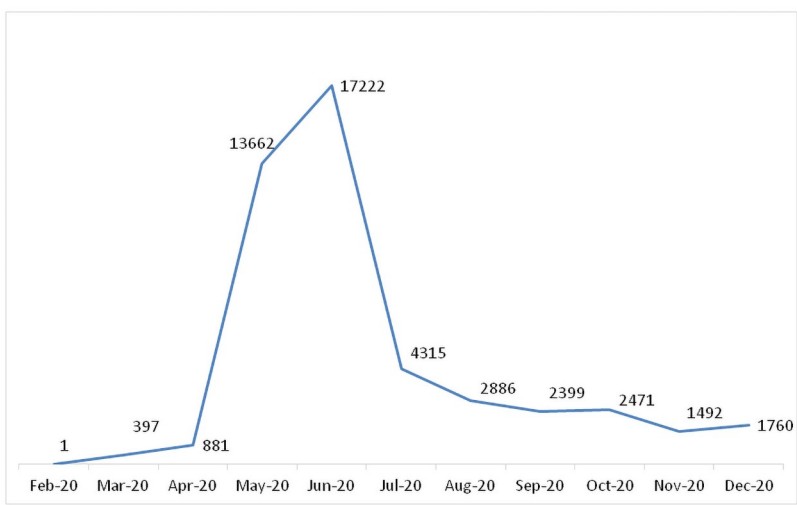

**Fig 6. Number of COVID-19 positive cases by month in Qatar in 2020.**

multisite health care system study from Mayo clinic in USA reported that 12.6% of the total vascular surgery encounters were completed via telemedicine between March 2020 and August 2020 [12]. Approximately 81% of these patients rated their overall healthcare experiences 'very good' and this rating was comparable to face-to-face encounters (79%) [12].

A cross-sectional survey by the Society for Vascular Surgery (SVS) among vascular surgeons in the US found that 91% of the surgeons in the survey reported the cancellation of all elective surgeries and 89% reported disruption to their outpatient clinics or ambulatory center schedules [13]. Few surgeons (8%) indicated that they performed elective cases focusing on dialysis access (59%), followed by aortic repair (51%), and lower extremity revascularization (49%) [13]. In our report, there was 97% decrease in procedures performed in the vascular clinics, and diabetic foot dressing was the main procedure (38%) during the COVID era. The SVS survey also revealed that 81% of the surgeons used telehealth services [13].

However, other studies showed that patients were more favorable towards utilization of telemedicine services, especially due to the COVID scenario. In a Chinese study, 95% of vascular patients favored telemedicine instead of postponing their appointments and all the patients expressed their satisfaction with the video calls and preferred to use telemedicine for follow-up in the future [14]. This helps vascular patients to avoid the probability of transmission and infection of COVID associated with travel, hospital visit and physical consultation [14]. It also helps to save personal protective equipment in hospitals and protect healthcare workers while minimizing the chances of spreading the virus [15]. Adopting such teleconsultation will improve the preparedness of different subspecialties for possible subsequent new waves of the pandemic.

Teleconsultation still has many challenges including reimbursement, accessibility, monitoring of quality, patient confidentiality, local and national legal oversight and liability, Internet speed, patients' educational and know-how computer literacy level [12, 16]. Although there are multiple reports on the utilization of telemedicine services in different specialties worldwide, there is a paucity of studies from the Middle East [17] and the unique social and legal parameters which could affect clinical best practice. It is important to note that the ability to continue the development of hybridized services is contingent on an understanding of social and legal operational contexts. The acceptance of virtual consultations may be preferable to

cancelation during a pandemic, however, that cannot be taken as a preference. Within the given cultural context, there may be many patients who are unfamiliar or uncomfortable with the use of virtual means that could include video or photography. The sharing of videos or photos, especially those of a personal or sensitive nature, is not a common practice. Within the in-person clinical setting there is an ability to build the trust and rapport that is required for accurate diagnosis and to overcome apprehensions. The relationship with the patient may not be as easily established virtually and may account for the difference in the number of new cases from 2019 to 2020.

Apprehension around telemedicine may also exist because the practice often outpaces the legal parameters in place and organizations need to be responsive in establishing their own policies and procedures that protect both the patients and the providers. Qatar introduced its first data protection law under Law 13 of 2016. This law was revisited in November 2020, 8 months after the confirmation of the first COVID-19 case in Qatar, to add 14 regulatory guidelines which were not previously covered including information around Special Nature Personal Data such as health data relating to COVID-19 [18]. However, new regulatory guidelines continue to put the onus on organizations to translate the regulations into application. For the telemedicine setting, the questions around data security and privacy are numerous and of varying degrees of gravity. Should all the calls be conducted on hospital-provided accounts and devices? How are the data stored and which channels are appropriate to utilize for sharing information? These questions represent only some of the issues that need to be taken into consideration when establishing a productive ecosystem for telemedicine. Furthermore, the development of audits, patient satisfaction surveys and studies on provider experience would provide crucial insight that would help to identifying patient needs and address any technical, social or educational barriers to effective hybridized care models.

Patient selection in telemedicine services, especially in vascular surgery remains crucial [15]. Postoperative patients having an unremarkable and uncomplicated convalescence, chronic vascular patients requiring surveillance with or without vascular imaging, established patients with new complaint before triaged towards to hospital and new referral patients to determine whether to be seen immediately, are well suited for teleconsultations [15]. The need to perform physical examinations of the vascular tree, wound probing, or to perform necessary procedures without delay such as wound debridement, drainage, and suture removal require hybrid system.

The concept of the Vascular Surgery COVID-19 Collaborative (VASCC), to obtain international prospective data on the impact of widespread vascular surgical care delays as result of a national crisis and pandemic is very promising to increase physician and public awareness and preparedness during new waves of the disease [19].

The main limitation of this study is lack of data on patient and provider satisfaction regarding use of telemedicine services. However, this is a unique study in the Middle East region that describes the outpatient vascular surgery clinic experience during the covid-19 pandemic to set the best patient care during such pandemic and to improve the learning curve in such unprecedented crisis. The sample in the present study is representative of the country population as all vascular surgery patients are treated at the HGH. Lastly, we are working on the follow up of the current study cohort to address the strength and weakness of the given practices.

## Conclusions

Advances in technology and the current pandemic situation have contributed to the transition in the mode of patient care access from the traditional in-person patient encounters to the use of telemedicine services in the vascular surgery department. Increased popularity in

telemedicine is evident from previous studies which described number of patient visits and patient-satisfaction surveys. In addition, health care providers are also in agreement with these factors and expressed their convenience in continuing use of telemedicine even in the post-COVID era. However, further studies are required to establish these norms and to understand the unique social and legal context and its potential impact on practice. It is likely that post-COVID era will adopt hybrid care which combines virtual and in-person experiences across the full continuum of care, and it is important that consideration is given to the establishment and review of validated practice guidelines.

## Acknowledgments

The authors thank all the staff of vascular surgery section.

## Author Contributions

**Conceptualization:** Hassan Al-Thani, Ahmed Hussain, Ahmed Sharaf, Ahmed Sadek, Anas Aldakhl-Allah, Ahmed Awad, Nassar Al-Abdullah, Ahmad Zitoun, Jini Paul, Pushpalatha Pillai, Sara John, Ayman El-Menyar.

**Data curation:** Hassan Al-Thani, Ahmed Hussain, Ahmed Sharaf, Ahmed Sadek, Anas Aldakhl-Allah, Ahmed Awad, Nassar Al-Abdullah, Ahmad Zitoun, Jini Paul, Pushpalatha Pillai, Sara John.

**Formal analysis:** Hassan Al-Thani, Ahmed Hussain, Ahmed Sharaf, Ahmed Sadek, Anas Aldakhl-Allah, Ahmed Awad, Nassar Al-Abdullah, Ahmad Zitoun, Jini Paul, Pushpalatha Pillai, Sara John.

**Methodology:** Ahammed Mekkodathil, Ayman El-Menyar.

**Writing – original draft:** Hassan Al-Thani, Ahammed Mekkodathil.

**Writing – review & editing:** Ayman El-Menyar.

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
