## [Decision Letter · Decision Letter 0]

30 Jun 2021

PONE-D-21-19489

Implementation of Vascular Surgery Teleconsultation during the COVID-19 Pandemic: Insights for Practice from a Tertiary Care Hospital in Qatar

PLOS ONE

Dear Dr. El-Menyar,

Thank you for submitting your manuscript to PLOS ONE. After careful consideration, we feel that it has merit but does not fully meet PLOS ONE’s publication criteria as it currently stands. Therefore, we invite you to submit a revised version of the manuscript that addresses the points raised during the review process.

Please address the issues and revise accordingly.

We look forward to receiving your revised manuscript.

Kind regards,

Academic Editor

PLOS ONE

Journal Requirements:

Reviewers' comments:

Reviewer's Responses to Questions

**Comments to the Author**

1. Is the manuscript technically sound, and do the data support the conclusions?

Reviewer #1: Partly

Reviewer #2: No

2. Has the statistical analysis been performed appropriately and rigorously? 

Reviewer #1: Yes

Reviewer #2: I Don't Know

3. Have the authors made all data underlying the findings in their manuscript fully available?

Reviewer #1: Yes

Reviewer #2: Yes

4. Is the manuscript presented in an intelligible fashion and written in standard English?

Reviewer #1: Yes

Reviewer #2: No

5. Review Comments to the Author

Reviewer #1: While not unique, the authors have done an excellent job at quantifying the effects of the covid pandemic on their outpatient practice and have highlighted the use of telemedicine during the pandemic and emphasized its ongoing utility after the pandemic.

1. Results, page 7: What was the indication for sclerotherapy? Even though decreased this was still the most common procedure. In most cases this is cosmetic and purely elective. Most places discontinued purely elective procedures during the height of the pandemic.

2. Do the authors have information on telephone vs computer/video visits? In our experience, not all patients have adequate computer resources for video visits, so telephone visits are necessary.

3. The authors focus on outpatient encounters. Do they have concomitant information on pandemic effects of inpatient encounters and surgical procedures?

Reviewer #2: Please see the attached file for detailed comments. I think the present study does have potential, but would benefit from altered statistical analyses of the data file. According to the manuscript all needed data should be available.

6. PLOS authors have the option to publish the peer review history of their article (what does this mean?). If published, this will include your full peer review and any attached files.

Reviewer #1: No

Reviewer #2: No

---

## [Author Response · Author response to Decision Letter 0]

13 Jul 2021

Reviewer #1: While not unique, the authors have done an excellent job at quantifying the effects of the covid pandemic on their outpatient practice and have highlighted the use of telemedicine during the pandemic and emphasized its ongoing utility after the pandemic.

1. Results, page 7: What was the indication for sclerotherapy? Even though decreased this was still the most common procedure. In most cases this is cosmetic and purely elective. Most places discontinued purely elective procedures during the height of the pandemic.

2. Do the authors have information on telephone vs computer/video visits? In our experience, not all patients have adequate computer resources for video visits, so telephone visits are necessary.

3. The authors focus on outpatient encounters. Do they have concomitant information on pandemic effects of inpatient encounters and surgical procedures?

Reply: 

- This is a unique study in our Middle East region that describes the outpatient vascular surgery clinic experience during the covid-19 pandemic which needs to be shared with others to set the best patient care during such pandemic crisis. Sclerotherapy procedure was stopped during the Covid pandemic as illustrated in the figure. (In the first three month 459 cases done and only one case of sclerotherapy in September for bleeding varicose vein).

- All the information based on telephone conversation. 

- During the pick of the pandemic there was a shift in the Multi-disciplinary outpatient vascular clinic for hemodialysis vascular access to inpatient consultation or direct consultation in the hemodialysis centers. This resulted in a significant drop in the number of physical visit to the vascular clinic for the hemodialysis patients. 

- On the other hand, vascular access for chemotherapy rarely came to the vascular clinic even before the pandemic. The vascular coordinator for chemotherapy access at the national oncology center (NCCR) referral done for new chemotherapy access or on going issues with patients need chemotherapy and patients seen at NCCR physically by the vascular team. 

- All the emergency vascular consultations are seen by the vascular surgeon on-call physically whether inpatient or in the emergency department. The pandemic did not impact on the overall number of consultation. On the other hand the surgical procedures decreased from 2506 in 2019 to 1849 in 2020 due to the discontinuation of elective varicose vein surgery. 

Reviewer #2: Please see the attached file for detailed comments. I think the present study does have potential, but would benefit from altered statistical analyses of the data file. According to the manuscript all needed data should be available. Statistical analyses especially table 2 should be presented more clearly in methods section and in the legend.

Reply: done thanks

Results table 2 is hard to read (p value chi test between which values?) more accurate presentation and legend would be helpful and provide sufficient information on what is presented. Figure 3 why not present year 2020 together with figure 5 on same panel? This would be more informative? It is not clear if authors present number of out-patient clinic or vascular department. Maybe mixed? State accurately what numbers stand for.

Reply: The current study present vascular surgery outpatient clinic cases only. Tables and figures revised as per reviewer request 

Discussion would be nice to begin with the major observations of the study. Second and third paragraph essential for the manuscript? If authors could demonstrate significant differences between urgent and elective vascular patient workload during pandemics and possibly these patient groups vary in requirements of out-patient contact (face-to-face or telecommunication)? Also, the multidisciplinary patient groups are interesting on this concept.

Reply: 

- Discussion revised as per reviewer request. 

- Our study demonstrated that 61% of the outpatient encounters at the vascular clinic was through teleconsultation in the year 2020 (COVID-19 era). Females were the majority of patients who accessed care in this period, and many of cases were follow-ups. Consultations for new vascular cases reduced significantly. The duration from February 2020 to April 2020 witnessed 95% drop in the physical visits of patients and more than 25 times increase in the telemedicine encounters. Similarly, there was 97% decrease in procedures performed in the vascular clinics during the COVID-19 period. During the pick of the pandemic, there was a shift in the multi-disciplinary outpatient vascular clinic for hemodialysis vascular access to inpatient consultation or direct consultation in the hemodialysis centers. This resulted in a significant drop in the number of physical visit to the vascular clinic for the hemodialysis patients. 

- On the other hand, patients for vascular access for chemotherapy rarely come to the vascular clinic even before the pandemic. The vascular coordinators for chemotherapy access at the national oncology center (NCCR) arrange referral for new chemotherapy access or ongoing issues with patients need chemotherapy and patients will be seen at the NCCR physically by the vascular team. 

- All the emergency vascular consultations are seen by the vascular surgeon on-call physically whether inpatient or in the emergency department. The pandemic did not impact on the overall number of consultation. On the other hand, the surgical procedures decreased from 2506 in 2019 to 1849 in 2020 due to the discontinuation of elective varicose vein surgery. Oncology-Vascular surgery Multidisciplinary Video Conferencing Meetings Calling using Microsoft Teams to support other surgical specialties for cancer cases take place twice-weekly ; one the uro-oncology Multidisciplinary and the other one in the sarcoma Multidisciplinary team.

- Diagnosis Category for the patients seen in the outpatient clinic is illustrated in figure 3&4. Figure 6 added

In the present manuscript authors have analyzed the frailty and benefit of telecommunication as tool for organizing vascular surgery outpatient clinic during Covid-19 era. This is a timely topic for the research and authors have a large cohort. Authors have a valid scope for the study “The goal of this study is to describe the utilization of telemedicine services in the vascular surgery clinic in a tertiary hospital in Qatar and explore some considerations for practice from the Qatar context that may be relevant in other settings globally” The major criticism is the grouping of this nice amount of valuable data, which can provide potentially interesting valuable new data on which patient sub-groups can be managed with telecommunication and who really require face-to-face contact to out-patient clinic. Comparing groups like nationals versus non-nationals does not provide globally relevant new data. The over 10 000 visits would provide more general importance if divided based on clinical presentation of vascular disease or requirement of vascular services (access patients etc). Comparing numbers of venous patients C1-3 versus C4-6 telecommunication versus face-to-face, similarly numbers treated and referred before and after Covid-19 pandemic, claudication versus CLI patients, AAA, internal consultations (nephrology, neurology) etc would provide more general new insight to utility of telemedicine in special patient groups and workload of vascular services during pandemic. This data should be available on authors data base?

Reply: thanks for these invaluable comments. As you said we mainly aimed at describing our experience from a developing country and to share it with others to improve our learning curve in such unprecedented crisis. We are working on the follow up of such subgroups as per your suggestion which will take long time and will be the scope of further publication. We added this point to the current study limitations

Abstract: 

Background. Last sentence not fluent and could be pasted to another paragraph?

Reply: Sentence moved to the introduction of the main manuscript

Objectives and methods should be more accurate.

Reply: thanks, done

Results is hard to follow, and it is impossible to know where authors present data and numbers from outpatient clinic and when numbers present the workload of vascular clinic. This is also major criticism for all sections of the manuscript.

Reply: we addressed clearly that these data are from outpatient clinics only 

Conclusions first sentence is on focus and as stated in the major criticism the present study has potential answering many important aspects of vascular out-patients clinic.

Reply: thanks 

Introduction is nice

Reply: thanks

Methods: please see above. The grouping of the data could be more relevant. Table 1 figures 1-2 could be more readable? Statistical analyses especially table 2 should be presented more clearly in methods section and in the legend.

Reply: tables and figures improved accordingly

---

## [Decision Letter · Decision Letter 1]

21 Jul 2021

PONE-D-21-19489R1

Implementation of Vascular Surgery Teleconsultation during the COVID-19 Pandemic: Insights from the Outpatient vascular Clinics in a Tertiary Care Hospital in Qatar

PLOS ONE

Dear Dr. El-Menyar,

Thank you for submitting your manuscript to PLOS ONE. After careful consideration, we feel that it has merit but does not fully meet PLOS ONE’s publication criteria as it currently stands. Therefore, we invite you to submit a revised version of the manuscript that addresses the points raised during the review process.

Please revise accordingly.

We look forward to receiving your revised manuscript.

Kind regards,

Academic Editor

PLOS ONE

Journal Requirements:

Reviewers' comments:

Reviewer's Responses to Questions

**Comments to the Author**

1. If the authors have adequately addressed your comments raised in a previous round of review and you feel that this manuscript is now acceptable for publication, you may indicate that here to bypass the “Comments to the Author” section, enter your conflict of interest statement in the “Confidential to Editor” section, and submit your "Accept" recommendation.

Reviewer #1: All comments have been addressed

Reviewer #2: All comments have been addressed

2. Is the manuscript technically sound, and do the data support the conclusions?

Reviewer #1: Yes

Reviewer #2: Yes

3. Has the statistical analysis been performed appropriately and rigorously? 

Reviewer #1: Yes

Reviewer #2: Yes

4. Have the authors made all data underlying the findings in their manuscript fully available?

Reviewer #1: Yes

Reviewer #2: Yes

5. Is the manuscript presented in an intelligible fashion and written in standard English?

Reviewer #1: No

Reviewer #2: Yes

6. Review Comments to the Author

Reviewer #1: (No Response)

Reviewer #2: I think the manuscript has improved after revision. My only comment concerns paragraph above conclusions ...covid-19 pandemic which needs to be shared with others... in the final manuscript the "needs to be shared with others" could be deleted? This applies to editor and reviewers?

7. PLOS authors have the option to publish the peer review history of their article (what does this mean?). If published, this will include your full peer review and any attached files.

Reviewer #1: No

Reviewer #2: No

---

## [Author Response · Author response to Decision Letter 1]

21 Jul 2021

Comments to the Author

1. If the authors have adequately addressed your comments raised in a previous round of review and you feel that this manuscript is now acceptable for publication, you may indicate that here to bypass the “Comments to the Author” section, enter your conflict of interest statement in the “Confidential to Editor” section, and submit your "Accept" recommendation.

Reviewer #1: All comments have been addressed

Reviewer #2: All comments have been addressed

Reply: thanks

2. Is the manuscript technically sound, and do the data support the conclusions?

Reviewer #1: Yes

Reviewer #2: Yes

Reply: thanks

3. Has the statistical analysis been performed appropriately and rigorously?

Reviewer #1: Yes

Reviewer #2: Yes

Reply: thanks

4. Have the authors made all data underlying the findings in their manuscript fully available?

Reviewer #1: Yes

Reviewer #2: Yes

Reply: thanks

5. Is the manuscript presented in an intelligible fashion and written in standard English?

Reviewer #1: No

Reviewer #2: Yes

Reply: thanks, Done

6. Review Comments to the Author

Reviewer #1: (No Response)

Reviewer #2: I think the manuscript has improved after revision. My only comment concerns paragraph above conclusions ...covid-19 pandemic which needs to be shared with others... in the final manuscript the "needs to be shared with others" could be deleted? This applies to editor and reviewers?

Reply: thanks, Done

7. PLOS authors have the option to publish the peer review history of their article (what does this mean?). If published, this will include your full peer review and any attached files.

Do you want your identity to be public for this peer review? For information about this choice, including consent withdrawal, please see our Privacy Policy.

Reviewer #1: No

Reviewer #2: No

Journal Requirements: Please review your reference list to ensure that it is complete and correct. If you have cited papers that have been retracted, please include the rationale for doing so in the manuscript text, or remove these references and replace them with relevant current references. Any changes to the reference list should be mentioned in the rebuttal letter that accompanies your revised manuscript. If you need to cite a retracted article, indicate the article’s retracted status in the References list and also include a citation and full reference for the retraction notice.

Reply: no retracted ref found

---

## [Decision Letter · Decision Letter 2]

18 Aug 2021

PONE-D-21-19489R2

Implementation of Vascular Surgery Teleconsultation during the COVID-19 Pandemic: Insights from the Outpatient vascular Clinics in a Tertiary Care Hospital in Qatar

PLOS ONE

Dear Dr. El-Menyar,

Thank you for submitting your manuscript to PLOS ONE. After careful consideration, we feel that it has merit but does not fully meet PLOS ONE’s publication criteria as it currently stands. Therefore, we invite you to submit a revised version of the manuscript that addresses the points raised during the review process.

Please address the issues and revise accordingly.

We look forward to receiving your revised manuscript.

Kind regards,

Academic Editor

PLOS ONE

Reviewers' comments:

Reviewer's Responses to Questions

**Comments to the Author**

1. If the authors have adequately addressed your comments raised in a previous round of review and you feel that this manuscript is now acceptable for publication, you may indicate that here to bypass the “Comments to the Author” section, enter your conflict of interest statement in the “Confidential to Editor” section, and submit your "Accept" recommendation.

Reviewer #2: All comments have been addressed

Reviewer #3: All comments have been addressed

Reviewer #4: All comments have been addressed

2. Is the manuscript technically sound, and do the data support the conclusions?

Reviewer #2: Yes

Reviewer #3: Yes

Reviewer #4: Yes

3. Has the statistical analysis been performed appropriately and rigorously? 

Reviewer #2: Yes

Reviewer #3: N/A

Reviewer #4: Yes

4. Have the authors made all data underlying the findings in their manuscript fully available?

Reviewer #2: Yes

Reviewer #3: Yes

Reviewer #4: Yes

5. Is the manuscript presented in an intelligible fashion and written in standard English?

Reviewer #2: Yes

Reviewer #3: Yes

Reviewer #4: Yes

6. Review Comments to the Author

Reviewer #2: All presented comments are addressed.

Reviewer #3: It is clear that a lot of hard work has gone in to preparing this manuscript, however it is my feeling that rather than generating new knowledge this is more of an audit/local service evaluation and perhaps could be usefully relayed in a short letter to the editor rather than a full manuscript.

Reviewer #4: Dear author,

Your submission is very well-written and was able to adress all suggested comments raised from the previous revision.

There is still a minor mistake at page 22: "Vascular surgery for arthrosclerosis" instead of atherosclerosis, which should be corrected.

7. PLOS authors have the option to publish the peer review history of their article (what does this mean?). If published, this will include your full peer review and any attached files.

Reviewer #2: No

Reviewer #3: No

Reviewer #4: No

---

## [Author Response · Author response to Decision Letter 2]

19 Aug 2021

Thanks for the reviewers and editor. There was only one comment that already has been addressed.

Comments to the Author

1. If the authors have adequately addressed your comments raised in a previous round of review and you feel that this manuscript is now acceptable for publication, you may indicate that here to bypass the “Comments to the Author” section, enter your conflict of interest statement in the “Confidential to Editor” section, and submit your "Accept" recommendation.

Reviewer #2: All comments have been addressed

Reviewer #3: All comments have been addressed

Reviewer #4: All comments have been addressed

2. Is the manuscript technically sound, and do the data support the conclusions?

Reviewer #2: Yes

Reviewer #3: Yes

Reviewer #4: Yes

3. Has the statistical analysis been performed appropriately and rigorously?

Reviewer #2: Yes

Reviewer #3: N/A

Reviewer #4: Yes

4. Have the authors made all data underlying the findings in their manuscript fully available?

Reviewer #2: Yes

Reviewer #3: Yes

Reviewer #4: Yes

5. Is the manuscript presented in an intelligible fashion and written in standard English?

Reviewer #2: Yes

Reviewer #3: Yes

Reviewer #4: Yes

6. Review Comments to the Author

Reviewer #2: All presented comments are addressed.

Reviewer #3: It is clear that a lot of hard work has gone in to preparing this manuscript, however it is my feeling that rather than generating new knowledge this is more of an audit/local service evaluation and perhaps could be usefully relayed in a short letter to the editor rather than a full manuscript.

Reply: Thanks, This is the first paper of this kind in the Middle East (and few worldwide) that addresses the importance and feasibility of teleconsultation in the field of vascular surgery during the unprecedented pandemic, therefore full manuscript is needed

Reviewer #4: Dear author,

Your submission is very well-written and was able to adress all suggested comments raised from the previous revision.

There is still a minor mistake at page 22: "Vascular surgery for arthrosclerosis" instead of atherosclerosis, which should be corrected.

Reply: thanks, corrected

7. PLOS authors have the option to publish the peer review history of their article (what does this mean?). If published, this will include your full peer review and any attached files.

Do you want your identity to be public for this peer review? For information about this choice, including consent withdrawal, please see our Privacy Policy.

Reviewer #2: No

Reviewer #3: No

Reviewer #4: No

---

## [Decision Letter · Decision Letter 3]

2 Sep 2021

Implementation of Vascular Surgery Teleconsultation during the COVID-19 Pandemic: Insights from the Outpatient vascular Clinics in a Tertiary Care Hospital in Qatar

PONE-D-21-19489R3

Dear Dr. El-Menyar,

We’re pleased to inform you that your manuscript has been judged scientifically suitable for publication and will be formally accepted for publication once it meets all outstanding technical requirements.

Kind regards,

Academic Editor

PLOS ONE

Additional Editor Comments (optional):

Reviewers' comments:

Reviewer's Responses to Questions

**Comments to the Author**

1. If the authors have adequately addressed your comments raised in a previous round of review and you feel that this manuscript is now acceptable for publication, you may indicate that here to bypass the “Comments to the Author” section, enter your conflict of interest statement in the “Confidential to Editor” section, and submit your "Accept" recommendation.

Reviewer #2: All comments have been addressed

Reviewer #4: All comments have been addressed

2. Is the manuscript technically sound, and do the data support the conclusions?

Reviewer #2: Yes

Reviewer #4: Yes

3. Has the statistical analysis been performed appropriately and rigorously? 

Reviewer #2: Yes

Reviewer #4: Yes

4. Have the authors made all data underlying the findings in their manuscript fully available?

Reviewer #2: Yes

Reviewer #4: Yes

5. Is the manuscript presented in an intelligible fashion and written in standard English?

Reviewer #2: Yes

Reviewer #4: Yes

6. Review Comments to the Author

Reviewer #2: All presented comments are addressed

Reviewer #4: Dear authors, thank you for submitting your review. There are not any other comments regarding your manuscript.

7. PLOS authors have the option to publish the peer review history of their article (what does this mean?). If published, this will include your full peer review and any attached files.

Reviewer #2: No

Reviewer #4: No

---

## [Editor Report · Acceptance letter]

17 Sep 2021

PONE-D-21-19489R3 

Implementation of Vascular Surgery Teleconsultation during the COVID-19 Pandemic: Insights from the Outpatient vascular Clinics in a Tertiary Care Hospital in Qatar 

Dear Dr. El-Menyar:

I'm pleased to inform you that your manuscript has been deemed suitable for publication in PLOS ONE. Congratulations! Your manuscript is now with our production department. 

Kind regards, 

on behalf of

Dr. Robert Jeenchen Chen 

Academic Editor

PLOS ONE